# A Study on the Wide Range of Relative Humidity in Cirrus Clouds Using Large-Ensemble Parcel Model Simulations

Miao Zhao and Xiangjun Shi *

School of Atmospheric Sciences, Nanjing University of Information Science and Technology, Nanjing 210044, China
* Correspondence: shixj@nuist.edu.cn

**Abstract:** This study investigates the possible mechanisms related to the wide range of relative humidity in cirrus clouds ($RH_i$). Under the closed adiabatic assumption, the impacts of vertical motion and ice crystal deposition/sublimation on $RH_i$ are investigated through in situ observations and parcel model simulations. Vertical motion is an active external force that changes the $RH_i$, and ice crystal deposition/sublimation plays a role in mitigating the change in the $RH_i$. They are the two most important mechanisms involved in controlling the $RH_i$ fluctuation during cirrus evolution and could well explain the wide range of $RH_i$ in wave-related cirrus clouds. Furthermore, a comparison of statistical cloud characteristics from both observations and simulations shows that a very low value (e.g., 0.001) for the water vapor ice deposition coefficient is highly unlikely.

**Keywords:** cirrus clouds; relative humidity; parcel model; large-ensemble simulations





## 1. Introduction

Cirrus clouds (including thin cirrus) cover about 30% of the Earth's surface and significantly impact the transport of water vapor across the tropopause and the radiative transfer in the upper troposphere [1–9]. Unlike in warm clouds, the number density ($N_i$) of ice crystals (ICs) in cirrus clouds is relatively low (usually much less than 1000 L$^{-1}$), and the in-cloud relative humidity over ice ($RH_i$) has a wide range (50% to 150% in most cases) [5,10–14]. The in-cloud $RH_i$ dominates the deposition/sublimation process of ICs and plays an important role in cloud evolution [15–17]. Although some observational studies have provided some possible reasons for the wide range of in-cloud $RH_i$, there are very few studies that quantitatively investigate the main mechanisms related to $RH_i$ [18–24].

The main influencing mechanisms of in-cloud $RH_i$ can be divided into two categories: mechanisms related to exchanges with the surrounding ambiance and mechanisms unassociated with the surrounding ambiance. The entrainment and mixing process on cloud edges can affect the in-cloud water through a mass exchange [25–27]. It is noteworthy that the entrainment and mixing process remains poorly understood and that its measurement and quantification are extremely difficult [28–31]. Thus, this study only focuses on the mechanisms unassociated with the surrounding ambiance. With the same water vapor pressure, the change in air temperature (T) caused by vertical motion could affect the $RH_i$ because the saturated water vapor pressure changes sharply with T [32–35]. In cirrus clouds, the IC deposition/sublimation process occurs as long as the water vapor is saturated/unsaturated [32]. The IC deposition/sublimation directly changes the in-cloud $RH_i$ via the changing water vapor. Meanwhile, the in-cloud $RH_i$ also determines the efficiency of the ICs to scavenge/release water vapor [16,36]. Taken overall, under conditions where neither a thermal exchange nor a mass exchange with the surrounding ambiance is possible (i.e., under the closed adiabatic assumption), the change in in-cloud $RH_i$ is mainly determined by vertical velocity (W) and IC deposition/sublimation. The main purpose of this study is to investigate how these two mechanisms impact $RH_i$ fluctuations and to

determine whether the wide range of $RH_i$ in wave-related cirrus clouds (this kind of cirrus is relatively more consistent with the closed adiabatic assumption) can be explained by these two mechanisms. Note that neither in-site observations nor the cloud microphysics classical theory can support this study alone.

The cloud parcel model is an important tool for analyzing cloud evolution [13,37–39]. Furthermore, field campaigns with aircraft measurements have provided numerous in situ observations of in-cloud $RH_i$, the ICs' physical characteristics, and the basic atmospheric state [24,40]. Based on these observations and a few artificial settings, the impact of vertical motion and IC deposition/sublimation on in-cloud $RH_i$ can be analyzed with large-ensemble parcel model simulations. In this study, a closed adiabatic cloud parcel model was first introduced and then used to carry out large-ensemble simulations. By taking advantage of the model simulations, the contributions of microphysical (i.e., IC deposition/sublimation) and dynamical (i.e., vertical motion) processes could also be separately investigated. This paper is organized as follows: Section 2 introduces the in situ observations and cloud parcel model, and the experimental setup is described. The parcel model simulation results are analyzed in Section 3. Section 4 discusses the limitations of this study. Finally, conclusions are provided in Section 5.

## 2. Data and Methods

### 2.1. In Situ Observations

Multiple field campaigns have provided the microphysical properties of cirrus clouds [24,40]. The preferred choice is that of cirrus clouds which are consistent with the closed adiabatic assumption. Cirrus clouds generally have two basic generating mechanisms, namely the uplift of moist air by wave motions on a variety of scales and the deep convective outflow [13,17]. The cirrus from the deep convective outflow grows thinner at a rapid pace as convection ceases, with IC sedimentation being an important process in their life cycle [41]. It is clear that this kind of cirrus is not suited for this study. In this study, in situ observations from the NASA Mid-latitude Airborne Cirrus Properties Experiment (MACPEX) were used because the cirrus observed during the MACPEX were mostly wave-related. The MACPEX mission was an intensive, month-long (March–April 2011) campaign over the South-Central U.S., and it provided nine flights with sufficient science-quality data [42,43]. Excluding the flights that targeted the anvil cirrus (i.e., deep convective outflow), seven flights were selected for the analysis of cirrus microphysical properties and the initial conditions for parcel model simulations.

The Meteorological Measurement System (MMS; [44]) provided the basic atmospheric states needed by this study, which included the W, T, and pressure (P). The water vapor was measured with a Harvard Water Vapor instrument (HWV; [45]). $RH_i$ was calculated from the measured water vapor pressure using the saturation water vapor pressure formulas of Murphy and Koop [46]. A Two-Dimensional Stereo probe (2D-S; [47]) provided the $N_i$, ice water content ($q_i$), and IC size distribution (the lower limit was 5 μm). The above-mentioned in situ observations were sampled at 1 Hz. Only the cirrus samples ($N_i > 1$ $L^{-1}$ and $T < 233$ K) that were far from the cloud edges were analyzed in this study. The number of total samples was 19,679.

### 2.2. Cloud Parcel Model and Experimental Setups

A closed adiabatic cloud parcel model driven by a prescribed W time series was used in this study. The P lapse rate of the surrounding ambiance can be diagnosed based on the standard atmosphere. The vertical motion will affect the P of the air parcel, then T, and then $RH_i$. The IC deposition/sublimation is described by the trend of the IC size ($R_i$, spherical assumption). Equations that describe the evolution of T, P, and $R_i$ can be found in the textbook by Pruppacher and Klett [32]. There are 50 IC size bins; in this study, $N_i$ and $q_i$ are the sums of each IC's size bin. $RH_i$ was determined using the conservation equation of total water. The IC sublimation was regarded as the reverse process of deposition. It is noteworthy that the impacts of vertical motion and IC deposition/sublimation on

$RH_i$ were well represented in the cloud parcel model because the relevant theoretical formulas are simple and robust. The parcel model was just used to investigate how these two mechanisms impact $RH_i$ fluctuations based on observed ICs, which would be used for the initial state of parcel model simulations. Besides the above mechanisms, only the ice nucleation process was added to the parcel model because the initial ice crystals might vanish in the sublimation phase during the simulation. The ice nucleation process included the heterogeneous freezing of ice nucleating particles (INPs) and the homogeneous nucleation of a supercooled aerosol solution. The residuals of the vanished ICs (excluding ICs from homogeneous freezing) would then 100% act as INPs during the subsequent simulation. The main purpose of this setup was to let the simulated $N_i$ have a statistical characteristic similar to that of the initial $N_i$ (i.e., observed $N_i$). The purpose of adding homogeneous freezing was to prevent the parcel model's simulated $RH_i$ from unreasonably high values (much larger than the homogeneous threshold, 150~160%). Following the simulation setup of Krämer et al. [39], the concentration of sulfate aerosol particles ($N_a$) (i.e., supercooled solution aerosol particles) used in the parcel model was assumed to be 300 cm$^{-3}$. It is necessary to point out that large-ensemble simulation results are not sensitive to $N_a$ because the concentration of newly formed ICs from homogeneous freezing ($N_{ihom}$) is not very sensitive to $N_a$ as long as the $N_a$ is much larger than the $N_{ihom}$ [38]. The model time step was set to 10 s. Correspondingly, the sample interval of the W time series used to drive the parcel model was 10 s.

In this study, the effects of vertical motion and IC deposition/sublimation were investigated using large-ensemble simulations, which could provide the statistical characteristics of cirrus clouds for a comparison with the observations. Each simulation ran for 6 h, which is around the lower limit of cirrus lifetime statistics (from several hours to more than 24 h [48]). The first-hour simulation was taken as the model spin-up, and the subsequent simulation results were used in our analysis. Because the main purpose of this study is to determine whether the wide range of $RH_i$ can be explained by the above two mechanisms, the observed data (especially IC and W) were used to constrain parcel model simulations. The initial state of the parcel model simulation was set by the observed T, P, $RH_i$, and IC size distribution (including the values of $N_i$ and $q_i$). Because the IC size bin in the parcel model (tunable setting) is usually different from that in the observation, the original data from the observed IC size distribution bin were first smoothed with the gamma size distribution before being used for the model's initial state. The observed W could not be directly used to drive the parcel model simulation because flight measurements were sampled from different places (~200 m between two adjacent samples). In other words, the observed W cannot provide wave frequency information. Therefore, only the amplitude of the W time series (i.e., vertical wind speed) used for driving the parcel model simulations came from observations. How to produce a reasonable W time series is carefully illustrated in the next paragraph.

During the parcel model simulation, the temperature perturbations (ΔTs, changes as compared to the initial value) were mostly determined based on the vertical distance of the cloud parcel from its initial position (ΔH, the time integration of W). Thus, the effects of vertical motion did not only depend on the wave amplitude but also on wave frequency. Here, the speed of vertical motion (i.e., wave amplitude) that was used to drive the parcel model was obtained from the observed W because the statistical distribution characteristic of the vertical wind speed at a local place had to be similar to that taken from flight measurements. The raw spectrum of vertical motion was generated by randomly selecting the W from the observation. In order to avoid producing a very weak ΔT time series, high-frequency turbulent waves (periods less than a few minutes) had to be inhibited to some extent. For this purpose, the value of the randomly selected W could not be far from the previous one (the difference was less than 0.5 m s$^{-1}$). To finally allow the cloud parcel to return to its initial position, the average W (very small) was removed. The ΔH was limited to −500~500 m in order to prevent an unusually strong ΔT (beyond the range of −5~5 K; the dry adiabatic lapse rate was at ~1 K/100 m). Furthermore, to allow the

ΔT time series to have obvious, alternating positive and negative variations during the 6-h simulation, a 6-h ΔH time series was required to have two peaks over 200 m and two valleys below −200 m. In this study, a total of 50 W time series (6-h series) were selected for the large-ensemble simulations. The statistical characteristics of these 50 W time series are shown in Figure 1. As expected, the occurrence frequency of W is very close to that in the observations. The ΔT is mostly dependent on the ΔH (i.e., time integration of W) under the closed adiabatic assumption [32]. In terms of statistical characteristics, the wave-driven ΔT is ~4 times that of the W. This relation between the ΔT and W is in accordance with the approximated formula ($\delta W = 0.23\,\delta T$; $\delta$ indicates a standard deviation) based on observed fluctuations on the mesoscale [36], which has been widely used in model simulation studies (e.g., [8,37,49,50]). The waves with periods of 20~200 min are dominant. This is generally in agreement with the vertical winds measured with the Doppler lidar at one mid-latitude place, which is dominated by mesoscale gravity waves (20~150 min, [51]). Figure 1 also shows two of the 50 W time series. The high-frequency fluctuations (period of a few minutes) due to turbulence are obvious. This is consistent with the vertical air motion retrieved from ground radar observations [38,52]. In short, the statistical characteristics of W and the corresponding ΔT are generally in agreement with related background knowledge, and so the prescribed W time series used for large-ensemble simulations are reasonable.

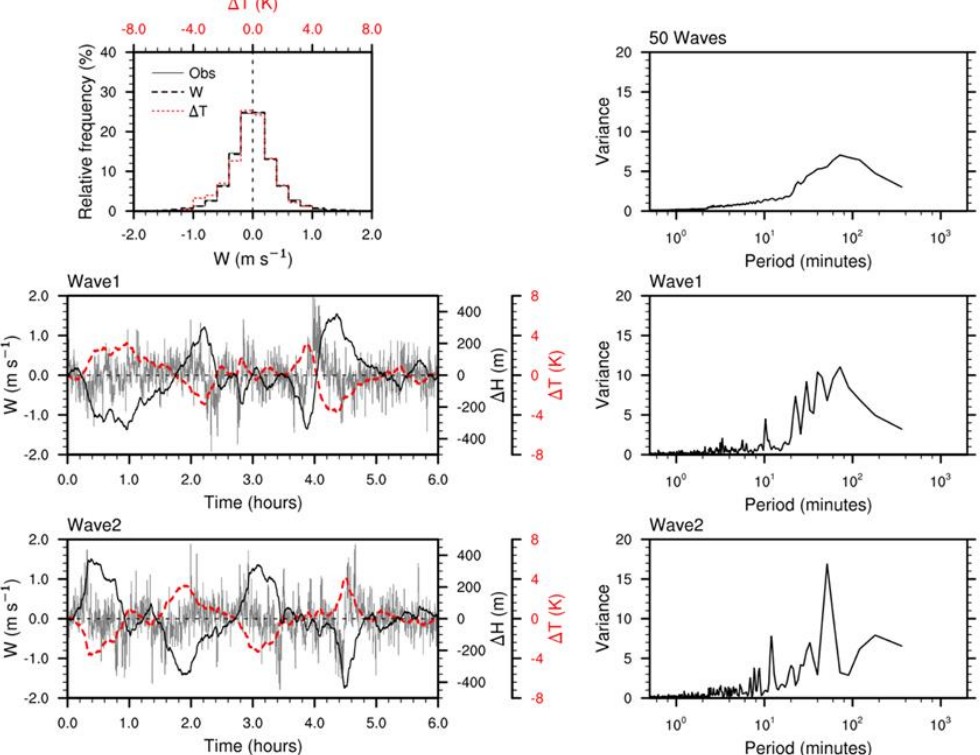

**Figure 1.** The statistical characteristics of the 50 prescribed W time series used to drive the parcel model (first row). They are the occurrence frequencies of W and ΔT (**left**) and the spectrum analysis of the W time series (**right**). The second and third rows present two cases (Wave1 and Wave2, respectively). The left columns show the time series of W (gray solid line), the ΔH (black solid line), and ΔT (red dotted line). The right columns show the spectrum analyses of the corresponding W time series. The ΔT and ΔH were calculated using the parcel model, with the initial conditions of T = 225 K and P = 250 hPa.

In this study, one thousand observed samples were used to derive the initial conditions needed for the large-ensemble simulation ($RH_i$, T, P, and ICs' variables). The probability distribution characteristics of these samples are consistent with those of the total samples (shown in Section 3.2). With each initial condition, the parcel model was

run 50 times, driven by the 50 W time series provided. Therefore, each experiment had 50,000 simulations (i.e., large-ensemble simulations). Here, one reference experiment and six sensitivity experiments were used to analyze the effects of vertical motion and IC deposition/sublimation (Table 1). The results of the REF experiment were compared with the observations and then investigated as to whether the statistical characteristics of the observations could be reproduced by parcel model simulations. Note that the water vapor ice deposition coefficient ($\alpha_d$), which determines the ability of ICs to scavenge/release water vapor, is full of uncertainty [53,54]. Gierens et al. [53] suggested that the $\alpha_d$ might be much smaller than a unit (i.e., 1). Thus, the $\alpha_d$ was set to 0.05 in the REF experiment. The Wamp, Wfre, and Wno experiments focused on investigating the effects of vertical motion. The Wamp experiment simulated cirrus evolution with a weaker vertical motion (where the amplitude was reduced by half). The Wfre experiment was used to test the impact of wave frequency. In order to double the wave frequency used in the Wfre experiment, the interval of the given W time series was set to 5 s rather than the initial 10 s. Correspondingly, each simulation was only run for 3 h. The Wno experiment simulated cirrus evolution without vertical motion. The effects of IC deposition/sublimation were investigated using the ICnosub, ICadL, and ICadH experiments. The comparison between the ICnosub and REF experiments was used to analyze the effect of the IC sublimation. The ICadL and ICadH experiments explored sensitivities to the $\alpha_d$. Because the efficiency of ICs in scavenging/releasing water vapor during the deposition/sublimation process depends on the value of the $\alpha_d$, the simulation results from the ICadL experiment (the $\alpha_d$ is very small) can be interpreted as cirrus evolution with a very weak impact from IC deposition/sublimation.

**Table 1.** The experiments carried out in this study.

| Experiment | Description |
|---|---|
| REF | Reference experiment. The $\alpha_d$ is 0.05. |
| Sensitivity experiments for vertical motion | |
| Wamp | Same as REF, but the amplitude of the W time series is reduced to half. |
| Wfre | Same as REF, but the frequency of the W time series is doubled. |
| Wno | Same as REF, but W has a constant value of 0. |
| Sensitivity experiments for the IC deposition/sublimation process. | |
| ICnosub | Same as REF, but the sublimation process is not allowed. |
| ICadH | Same as REF, but the $\alpha_d$ is set to 1.0. |
| ICadL | Same as REF, but the $\alpha_d$ is set to 0.001. |

## 3. Results

### 3.1. Case Studies

To better understand the cirrus statistical characteristics of large-ensemble simulations, the impacts of the vertical motion and microphysical processes on cirrus evolution are first illustrated with the use of some case simulations. The first W time series, shown in Figure 1 (Wave1), was used to drive seven simulations, which correspond to the seven experiments listed in Table 1. All these seven simulations had the same initial conditions: $RH_i$ = 110%, T = 225 K, P = 250 hPa, $N_i$ = 100 $L^{-1}$, the IC mass-weighted average radius ($R_{iq}$)= 15 μm, and the shape parameter of the gamma IC size distribution ($\mu_i$) was 5. Besides the seven simulations, the REF experiment was also conducted again, with initial conditions of $N_i$ = 10 $L^{-1}$ and $R_{iq}$ = 30 μm and was driven by another W time series (Wave2 in Figure 1). Thus, there were a total of eight case simulations.

Figure 2 shows the results of these case simulations. At the beginning of the REF (Wave1) simulation (0~0.2 h), all ICs grew because the $RH_i$ > 100%. As a result, the $q_i$ increased, and the $RH_i$ decreased. Thereafter, the ice parcel went down (0.3~1 h; see the $\Delta H$ in Figure 1). The $RH_i$ dropped to below 100%, mainly due to the positive $\Delta T$ driven by the $\Delta H$ (a negative correlation between the $\Delta T$ and $\Delta H$). Meanwhile, the ICs began to sublimate.

All ICs became smaller, and the small ICs vanished. Correspondingly, $N_i$ dropped from $100\ L^{-1}$ to $3\ L^{-1}$. At about 1.8 h, the increasing $RH_i$ reached the heterogeneous freezing threshold (120~130%). The heterogeneous freezing occurred with the aid of the INPs, which came from the sublimated ICs. Correspondingly, $N_i$ went back to $100\ L^{-1}$. In the subsequent simulation process, $N_i$ hardly changed. The size of each IC increased or decreased according to the $RH_i$ fluctuations, which were mainly driven by the $\Delta T$. It is noteworthy that $\Delta T$ is mainly dominated by the $\Delta H$ (i.e., vertical motion) because the effect of latent heat from IC deposition/sublimation is much weaker than the $\Delta H$ (not shown). In terms of water vapor, IC deposition/sublimation (i.e., the change in $q_i$) could directly change the $RH_i$ (the total water conservation) and then pull the $RH_i$ towards 100% (negative feedback). Taken overall, vertical motion can be considered an active external force that changes the $RH_i$, while IC deposition/sublimation plays a role in mitigating the change in $RH_i$.

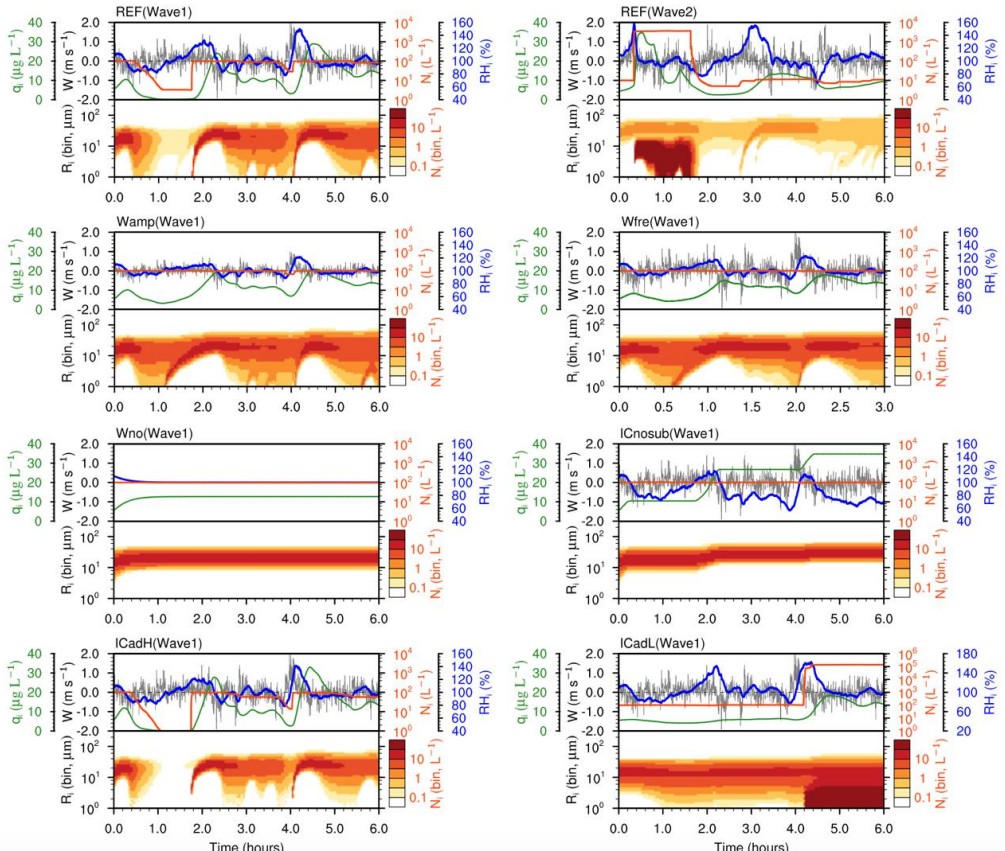

**Figure 2.** Cirrus evolutions from parcel model simulations using the W time series shown in Figure 1 (Wave1 or Wave2). Shown are $RH_i$ (blue thick solid line), W (gray thin solid line), $N_i$ (orange thin solid line), $q_i$ (green thin solid line). Black dotted line represents the line of W = 0, also the line of $RH_i$ = 100%. The shaded figures indicate the $N_i$ contribution from each radius bin. A total of 50 bins were used in this study. Experiment names and the corresponding W time series are marked in the upper left corners. Note that the ranges of the $N_i$-axis and $RH_i$-axis from the ICadL experiment are different from the others.

At the beginning of the REF (Wave2) simulation (Figure 2), the $RH_i$ increased quickly and reached the homogeneous freezing threshold (150~160%). Homogeneous nucleation occurred and produced a lot of small ICs. Because the newly formed ICs (whose concentration was relatively high) grew by scavenging water vapor, the $RH_i$ sharply fell to 100%. In the subsequent simulation, these small ICs would be the first to vanish during the sublimation phase (i.e., $RH_i$ below 100%). This suggests that it is difficult for the small

ICs produced by homogeneous freezing to survive for a long time if there are obvious vertical fluctuations.

As expected, both the Wamp and Wfre simulations showed that the maximum $\Delta H$ decreased as compared with the REF (Wave1) simulation (not shown). Therefore, the $RH_i$ fluctuations in the Wamp and Wfre simulations also became weak, and all ICs could survive during the sublimation phase (Figure 2). Furthermore, the maximum $q_i$ from the Wamp and Wfre simulations was much lower than that from the REF experiment. This suggests that stronger $RH_i$ fluctuations could help ICs to grow bigger (i.e., a larger $R_i$). The above phenomenon is clearer in the Wno simulation. Because of the IC deposition, the initial $RH_i$ (110%) was reduced to 100% in the first half hour of the Wno simulation. It is reasonable to infer that the $RH_i$ after 1 h of simulation is almost not related to the initial $RH_i$. This is the reason why the first-hour simulation results are not used in the following statistical analysis. In short, vertical motion (i.e., external forcing) controls the change in $RH_i$ and determines cirrus evolution to some extent.

During the ICnosub simulation, all ICs could grow under $RH_i > 100\%$ but could not sublimate under $RH_i < 100\%$ (Figure 2). Correspondingly, the $q_i$ could only increase. As a result, the $RH_i$ gradually decreased during the air parcel vertical fluctuation because more and more water vapor was absorbed by the ICs when $RH_i > 100\%$. The ICadL and ICadH simulations explored the sensitivity to the $\alpha_d$. As mentioned above, IC deposition/sublimation could dampen the fluctuation of $RH_i$ (negative feedback). This effect increased/decreased with an increasing/decreasing the $\alpha_d$. During the first sublimation phase of the ICadH simulation, all ICs vanished and released water vapor to prevent the $RH_i$ from decreasing. As compared to the REF (Wave1) simulation, the fluctuation of the $RH_i$ weakened in the ICadH simulation due to the stronger efficiency of the IC deposition/sublimation. On the contrary, the fluctuation of the $RH_i$ became stronger in the ICadL simulation. Note that a strong fluctuation could allow the $RH_i$ to reach the homogeneous freezing threshold. After the occurrence of homogeneous nucleation, the huge number of ICs would bring the $RH_i$ down to 100% due to their strong efficiency in scavenging water vapor. The comparison among the ICadL, ICadH, and REF (Wave1) simulations clearly indicates that IC deposition/sublimation could dampen the fluctuation of $RH_i$ and make an impact on cirrus evolution.

*3.2. Comparisons between the REF Experiment and Observations*

This subsection illustrates the comparisons between the REF experiment (50,000 simulations) and observations (19,679 samples). It is noteworthy that the simulation results with $N_i < 1 \, L^{-1}$ (only a very small part) were excluded from the statistical analysis, which is consistent with the observed samples. Furthermore, large-ensemble simulations provide very large samples (i.e., model output time points). Therefore, only a small part (200,000) of those samples (randomly selected) was used to create the plots below. It is necessary to point out that the probability distribution characteristic of these 200,000 samples is consistent with that of the total samples (not shown). The efficiency of ICs to scavenge/release water vapor is proportional to $N_i R_i$ [55,56]. Here, $N_i R_i$ is the sum of that from each IC's size bin. In the following analysis, $N_i R_i$ is used to indicate the effect of IC deposition/sublimation on $RH_i$.

Figure 3 shows scatter plots of the W, $RH_i$, and $N_i R_i$. As expected, in the scatter plot of $RH_i$ vs. $N_i R_i$, the 1000 samples selected for parcel model simulations have similar probability distribution characteristics with all observed samples. The scatter plot of $RH_i$ vs. W from the observations shows that the number of points (i.e., the samples defined by the $RH_i$ and W) in the first ($RH_i > 100\%$, and $W > 0 \, m \, s^{-1}$) and third ($RH_i < 100\%$, and $W < 0 \, m \, s^{-1}$) quadrants is larger than that in the second ($RH_i > 100\%$, and $W < 0 \, m \, s^{-1}$) and fourth ($RH_i < 100\%$, and $W > 0 \, m \, s^{-1}$). This might be explained by the fact that the $RH_i$ increased when the air parcel rose (i.e., $W > 0 \, m \, s^{-1}$, and T is decreased), and the $RH_i$ decreased when the air parcel went down (i.e., $W < 0 \, m \, s^{-1}$, and T is increased). This characteristic also appears in the corresponding scatter plot from the REF experiment. The observation data show that the number of samples with a higher $N_i R_i$ ($>1000 \, \mu m \, L^{-1}$)

when $RH_i > 125\%$ is obviously larger than that when $RH_i < 75\%$. The reason is that the $R_i$ begins to increase when $RH_i > 100\%$, and $N_i$ could possibly increase when $RH_i > 125\%$ (if ice nucleation occurs). When $RH_i < 75\%$, ICs will have been sublimated for a while ($R_i$ is decreased), and small ICs might vanish ($N_i$ decreases). On the other hand, the number of samples with lower $N_iR_i$ ($<100$ μm $L^{-1}$) when $RH_i < 75\%$ is obviously larger than that when $RH_i > 125\%$. Simulation results from the REF experiment also show these characteristics. In terms of the ICs' impact on $RH_i$, a higher $N_iR_i$ indicates that the $RH_i$ could be quickly pulled back to 100%. Therefore, the $RH_i$ is much closer to 100% when $N_iR_i > 10,000$ μm $L^{-1}$. This phenomenon is more obvious in the REF experiment. The scatter plots of W vs. $N_iR_i$ from both the observations and the REF experiment show that a higher $N_iR_i$ usually goes together with a stronger fluctuation range for the W. One possible reason is that the stronger W fluctuations could produce stronger $RH_i$ fluctuations, which might trigger ice nucleation events (i.e., $N_i$ is increased). Another possible reason is that ICs could grow bigger (i.e., a larger $R_i$ and $q_i$) under stronger W fluctuations (see the case simulations). Generally speaking, these scatter plots from the REF experiment are similar to those from the observations.

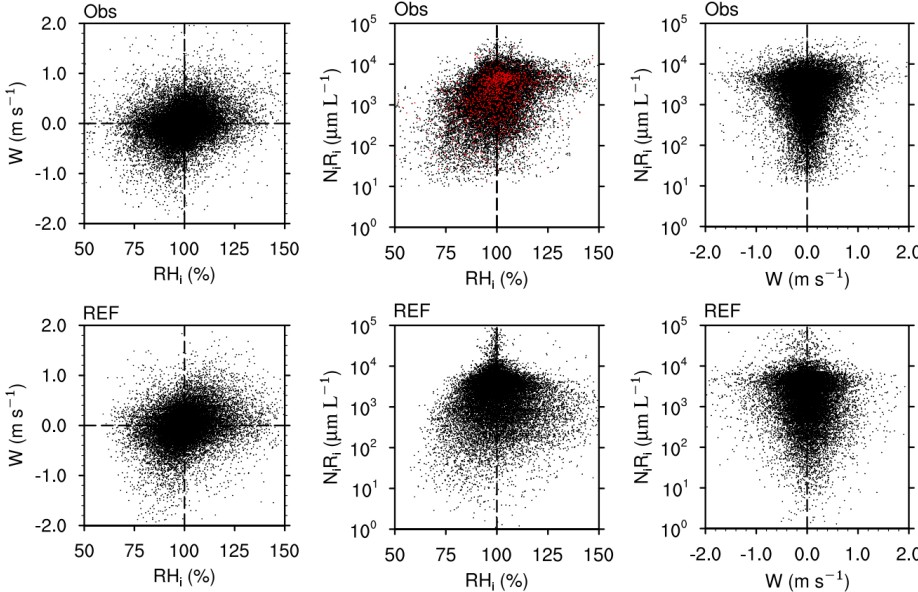

**Figure 3.** The scatter plots of W, $RH_i$, and $N_iR_i$ from observations (**upper panel**) and the REF experiment (**lower panel**). The dotted lines indicate $RH_i = 100\%$ or $W = 0$ m $s^{-1}$. Note that the 1000 samples selected for parcel model simulations are marked by red dots.

Figure 4 shows the occurrence frequencies of $RH_i$ and $N_iR_i$. The $RH_i$ from the observations is mostly in the range of 50~150%. This is generally in agreement with previous studies [12,17,18]. The occurrence frequency of $RH_i$ from the REF experiment is very close to that from the observations. Furthermore, the occurrence frequency of $N_iR_i$ from the REF experiment is also very close to that from the observations. Taken overall, the statistical characteristics of the W, $RH_i$, and $N_iR_i$ from the observations (Figures 1, 3 and 4) could almost be reproduced by the REF experiment (the large-ensemble simulations). This suggests that the wide range of $RH_i$ in the observations (wave-related cirrus clouds) can be well explained by the vertical motion and IC deposition/sublimation (two mechanisms under the closed adiabatic assumption).

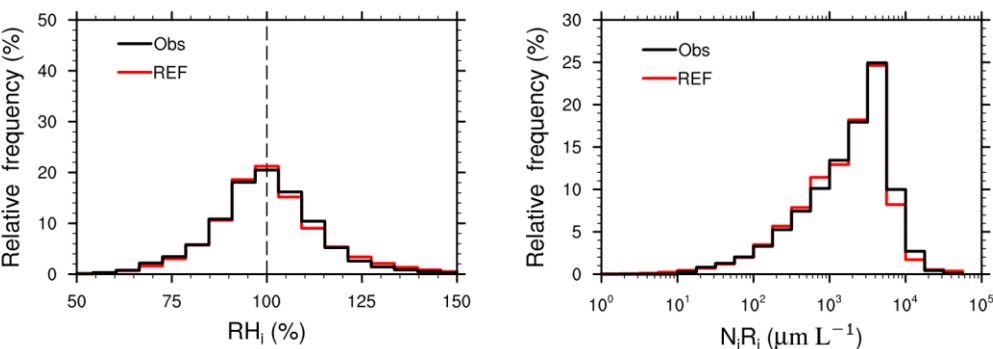

**Figure 4.** The occurrence frequencies of $N_iR_i$ and $RH_i$ from observations (black lines) and the REF experiment (red lines). The dotted line indicates $RH_i = 100\%$.

### 3.3. Deep Analysis through Sensitivity Experiments

In this subsection, the effects of the dynamical (vertical motion) and microphysical (IC deposition/sublimation) processes on $RH_i$ fluctuations are further investigated through several sensitivity experiments. It is noteworthy that results from the Wno experiment are not shown in the figure below (i.e., Figure 5) because it is difficult to illustrate the special statistical characteristics in figures (almost all samples have $RH_i = 100\%$, similar to the corresponding case simulation in Figure 2). Furthermore, unless otherwise specified, the results of the sensitivity experiments are analyzed and compared with those of the REF experiment.

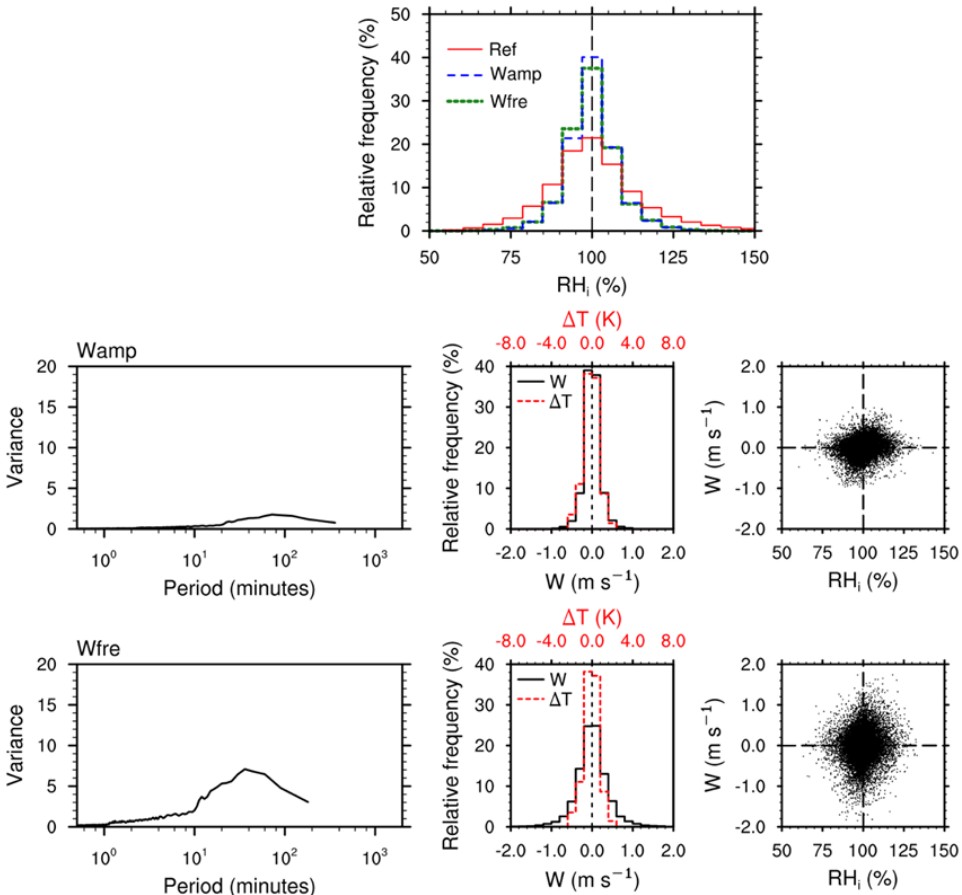

**Figure 5.** The occurrence frequency of $RH_i$ from the REF, Wamp, and Wfre experiments (**upper panel**). The lower panel shows the spectrum analysis of the W time series (**left**), the distributions of W and $\Delta T$ (**middle**), and the scatter plots of $RH_i$ vs. W (**right**) from the Wamp (**second row**) and Wfre (**third row**) experiments.

Without the vertical motion (i.e., the Wno experiment), the initial ICs scavenge/release water vapor, and then, in most cases, the $RH_i$ is pulled to 100% after the first hour (not shown), except for a few simulations with a much lower initial $RH_i$ (e.g., $RH_i$ = 60%) and a very small initial $q_i$ (i.e., ICs cannot provide enough water vapor). If $RH_i$ = 100%, there are no changes in the air parcel (i.e., cirrus evolution is stopped). This, again, suggests that vertical motion (i.e., external forcing) controls the change in $RH_i$, as well as the cirrus evolution. After reducing the amplitude of the vertical motion by half (i.e., the Wamp experiment), the occurrence frequency distribution of W becomes narrow (Figure 5). The occurrence frequency distribution of the $\Delta T$ also becomes narrow due to the reduced W. The fluctuation of $RH_i$ is mostly determined by the $\Delta T$. Therefore, the points ($RH_i$ and W) in the scatter plots (Figure 5) converge more towards the central point ($RH_i$ = 100%, and W = 0 m s$^{-1}$). After doubling the wave frequency of the vertical motion (i.e., the Wfre experiment), the curve of the spectrum analysis moves towards shorter periods (Figure 5). As expected, the $\Delta H$ is reduced by half (not shown). Therefore, the occurrence frequency distribution of $\Delta T$ becomes narrow, while the occurrence frequency distribution of the W is almost unchanged (Figure 5). Here, the relation between $\Delta T$ and the W is not consistent with the approximated formula ($\delta W = 0.23\ \delta T$) introduced in Section 2.2. This suggests that the wave spectrum characteristic used in the Wfre experiment is not consistent with reality. Furthermore, the points ($RH_i$ and W) in the scatter plot converge more towards the line of $RH_i$ = 100% (Figure 5). Although the occurrence frequency of the W is similar to that in the observations, the scatter plot of $RH_i$ vs. W is obviously different from the observations. This also suggests that the wave spectrum characteristic used in the Wfre experiment might not conform to reality. Both the Wamp and Wfre experiments show that the $RH_i$ is mainly within the range of 75~125% rather than 50~150% (Figure 5). The main reason for the narrow range of $RH_i$ is that the fluctuation of the $\Delta T$, which is mostly caused by the W, is weakened. In short, both the amplitude and frequency of the vertical motion determine the change in $RH_i$ during cirrus evolution.

Figure 6 shows the results of the sensitivity experiments for the IC deposition/sublimation process. In the scatter plots (both $RH_i$ vs. W and $RH_i$ vs. $N_iR_i$) from the ICnosub experiment, most points are at an $RH_i$ < 100% because the ICs can only scavenge water vapor. The scatter plot of $RH_i$ vs. $N_iR_i$ shows that the samples with a higher $N_iR_i$ (> 100 μm L$^{-1}$) are mostly from $RH_i$ < 75%. The reason is that a higher $N_iR_i$ usually indicates that most water vapor has been absorbed by the ICs. Because the ICs grew bigger (i.e., a larger $R_i$) during the simulation without IC sublimation (see the case simulations), it is more obvious in the scatter plot of W vs. $N_iR_i$ from the ICnosub experiment that a higher $N_iR_i$ usually goes with a wider range of W. Although the relative occurrence frequency of $RH_i$ at around 100% from the ICadH experiment is obviously larger than that from the REF experiments, the three scatter plots from the ICadH experiment do not show any obvious differences. However, the differences between the scatter plots from the ICadL and REF experiments are obvious. Because the effect of IC deposition/sublimation is weakened, the range of $RH_i$ becomes wider in the scatter plots of $RH_i$ vs. W. The scatter plots of both $RH_i$ vs. $N_iR_i$ and W vs. $N_iR_i$ show a big increase in the number of samples with a higher $N_iR_i$ (> 10,000 μm L$^{-1}$). This can be explained by the fact that homogeneous nucleation occurs (a large number of ICs is produced) more frequently due to a stronger $RH_i$ fluctuation. Because the efficiency of IC growth (i.e., the increase in $R_i$) is reduced, the phenomenon where a higher $N_iR_i$ (excluding $N_iR_i$ > 10,000 μm L$^{-1}$) goes with a wider range of W becomes weak in the scatter plot of W vs. $N_iR_i$. As compared to the ICnosub and ICadH experiments, the number of lower $N_iR_i$ (<10 μm L$^{-1}$) is obviously increased in the ICadL experiment because smaller ICs could survive longer. In short, it is clear that IC deposition/sublimation can significantly impact $RH_i$ as well as cirrus evolution.

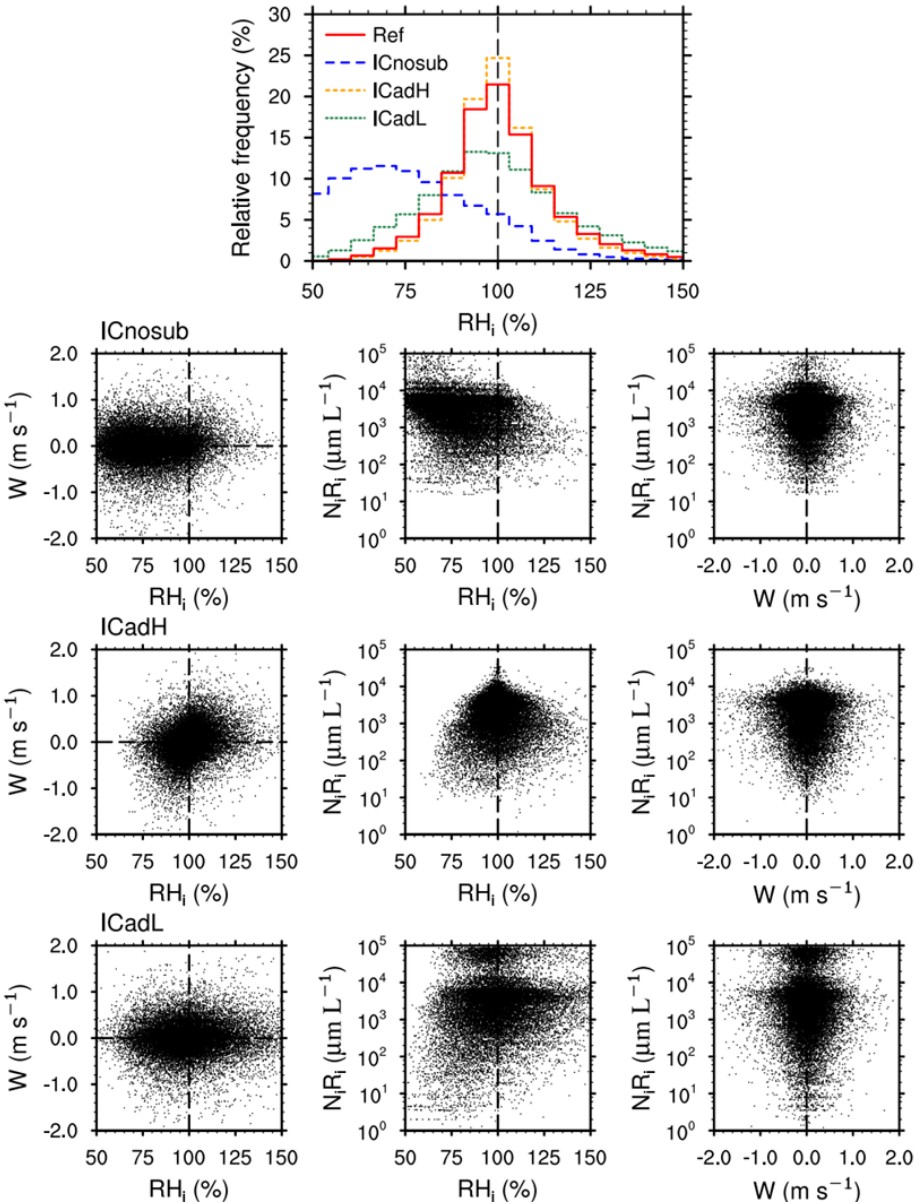

**Figure 6.** The occurrence frequency of $RH_i$ from the Ref, ICnosub, ICadH, and ICadL experiments (**upper panel**). The lower panel shows scatter plots similar to those in Figure 4 but for the ICnosub (**second row**), ICadH (**third row**), and ICadL (**fourth row**) experiments.

## 4. Discussion

It is necessary to discuss the uncertainty of the observation data. The accuracy of $RH_i$ is 10~20% [39], while the precision of the W measurements is about 0.05 m s$^{-1}$ [57]. Considering errors in measurement, the range of both W and $RH_i$ showed in this study might be slightly wider as compared with reality. Note that if the range of observed W and $RH_i$ (used for the parcel model initial state) becomes narrow, the range of simulated W and $RH_i$ also becomes narrow. The statistical characteristics of W and $RH_i$ from the observations could also be reproduced by the REF experiment (not shown). Because the ICs could be shattered during measurement, it is possible for $N_i$ to be overestimated [13,44,58]. Under the same $q_i$, $N_i R_i$ is decreased with decreasing $N_i$. An increase in the $\alpha_d$ (tunable parameter) can compensate for the impact of a decrease in $N_i R_i$ (i.e., weaker efficiency of ICs scavenging/releasing water vapor). In short, the uncertainty of the observation data cannot change the conclusion of this study (i.e., the wide range of observed $RH_i$ can be well explained by the vertical motion and IC deposition/sublimation).

One should keep in mind that this study was carried out based on the closed adiabatic assumption and that only wave-related cirrus clouds were selected for analysis. In order to examine the possibility of extending the conclusion beyond the closed adiabatic assumption, we also analyzed the observation data for convective cirrus clouds (not only the MACPEX but also another field campaign) and corresponding large-ensemble simulations. The statistical characteristics of the observations could not be well reproduced by the large-ensemble simulations. However, these large-ensemble simulations still suggest that both vertical motion and IC deposition/sublimation are two important mechanisms for determining the $RH_i$ fluctuation.

## 5. Conclusions

This study investigated the possible main mechanisms of the wide range of $RH_i$ in cirrus clouds. Considering the current measurement and theoretical limitations of the entrainment and mixing process on cloud edges [28–31], only the vertical motion and IC deposition/sublimation related to in-cloud $RH_i$ (two mechanisms under the closed adiabatic assumption) were investigated through in situ observations and parcel model simulations. The adiabatic closed cloud parcel model used in this study was driven by a prescribed vertical motion. The mostly wave-related cirrus observations from the MACPEX were used to analyze cloud microphysical properties and carry out parcel model simulations.

The results of the parcel model case simulations showed that vertical motion is an active external force that changes the $RH_i$, while IC deposition/sublimation plays a role in mitigating the change in the $RH_i$. These can be clearly illustrated by the cloud microphysics theory. The results from the large-ensemble parcel model simulations almost reproduced the statistical cloud characteristics from the observations. This suggests that vertical motion and IC deposition/sublimation are two of the most important mechanisms involved in controlling the $RH_i$ fluctuation and could well explain the wide range of $RH_i$ in wave-related cirrus.

There is an additional finding in this study. The comparison of some statistical characteristics from the observation and simulation results could be used to constrain some uncertain factors that determine cloud evolution but are difficult to measure. For instance, the wave frequency of the vertical motion could significantly impact the statistical characteristics in the $RH_i$ vs. W scatter plot. Conversely, the statistical characteristics in the $RH_i$ vs. W scatter plot could also be used to infer the statistical characteristics of wave frequency. The $\alpha_d$ is another example. The comparisons of the scatter plots of the W, $RH_i$, and $N_iR_i$ show that a very low value ($\alpha_d = 0.001$) for the $\alpha_d$ is highly unlikely.

**Author Contributions:** X.S. designed this study and wrote the Fortran code of the air parcel model. M.Z. carried out the simulations used in this study and created all figures. X.S. and M.Z. wrote the manuscript. All authors have read and agreed to the published version of the manuscript.

**Funding:** This research was funded by the National Key Research and Development Program of China (grant nos. 2018YFC1507001 and 2017YFA0604001) and the National Natural Science Foundation of China (grant nos. 41775095 and 42075145). The APC was supported by the same funders.

**Institutional Review Board Statement:** Not applicable.

**Informed Consent Statement:** Not applicable.

**Data Availability Statement:** The MAPEX observation data can be downloaded from the Earth Science Project Office (ESPO) https://espoarchive.nasa.gov/archive/browse/ (accessed on 6 December 2022). The Fortran code for the parcel model, the simulation results, and the NCL scripts have been archived in a public repository https://doi.org/10.5281/zenodo.7589993 (accessed on 31 January 2023).

**Acknowledgments:** This study was conducted at the High-Performance Computing Center of Nanjing University of Information Science and Technology.

**Conflicts of Interest:** The authors declare no conflict of interest.

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
