# Peer review of "A Study on the Wide Range of Relative Humidity in Cirrus Clouds Using Large-Ensemble Parcel Model Simulations"

_atmosphere, doi:10.3390/atmos14030583_

Round 1

Reviewer 1 Report

The results and conclusions of the study depend on some technical aspects of the parcel model simulations.

For instance, whether the ice nucleating particles (INPs) have a size distribution or a range of active site density. My impression is that the INPs are treated identically in the model. Also it would be helpful to know whether the results depend on the time step. Furthermore, the results also depend on initial ice crystal number concentration.

A single value seems overly simplified.

Reviewer 2 Report

Review of: A study on the wide range of relative humidity in cirrus clouds using large-ensemble parcel model simulations

By: Miao Zhao and Xiangjun Shi

Recommendation: Minor revision

Overview

This study investigated the effects of vertical motion and ice crystal deposition/sublimation on the changes in relative humidity in cirrus clouds. The authors conducted large number of numerical experiments and provided detailed analysis on the possible mechanisms related to the wide range of relative humidity. I have a few questions about the analysis of this study, which should be remedied prior to publication.

General comments

1.      The authors discussed different roles of homogeneous and heterogeneous nucleation of ice crystals with variation of relative humidity induced by different setting of vertical velocity and deposition/sublimation process. But it might be necessary to introduce the environmental conditions (e.g., temperature, relative humidity) that favor for different ice nucleation mechanism occurs.

Specific comments:

1.      For the introduction section, the authors are suggested to present the research progresses of main influencing mechanisms of in-cloud relative humidity in more detail, and present the innovativeness of the present study more clearly.

2.      Section 2.2: The present study employed a “closed adiabatic” cloud parcel model. What does “closed adiabatic” mean? Can the authors introduce the type of hydrometeors considered in this parcel model? Is there any other microphysical processes beside ice crystal deposition/sublimation?

3.      Figure 2: Why did not the number of ice crystal (Ni) change during most time of simulation period? Similar as the previous question, does the model consider collision-coalescence among ice particles?

Reviewer 3 Report

Comments on “A study on the wide range of relative humidity in cirrus clouds using large-ensemble parcel model simulations” by Zhao and Shi

The authors use a cloud parcel model to explore the impacts of vertical motion and ice microphysics (deposition and sublimation) on ice relative humidity. The model is initialized based on NASA Mid-latitude Airborne Cirrus Properties Experiment observations and a total of 50k simulations are conducted. Simulation results are then compared with in-situ observations. Results show that RHi fluctuations are controlled by vertical velocity and ice crystal deposition/sublimation in wave-related cirrus clouds. In general, the outcome of this study might be interesting to the community. However, some methods and results in this study are not clearly discussed. My major comments are listed below.

1.       Some details of the parcel model are missing. Since there is no page limit in this journal, I suggest the author add more details about the model setup such that the reader can better understand the results.

1)      Add more details about how homogeneous ice nucleation is treated in the model. Do the supercooled solution aerosol particles have the same size? When homogeneous freezing occurs, what is the initial size of ice particles?

2)      Does the model consider the stochastic ice nucleation process? When homogeneous freezing occurs, do all Na become ice particles, Nihom? If yes, I’m confused by the statement “It is necessary to point out that large-ensemble simulation results are not sensitive to Na because the concentration of newly formed ICs from homogeneous freezing (Nihom) is not sensitive to Na as long as Na is much larger than Nihom”. If not, it means that new ICs might form after each time step due to the stochastic ice nucleation process, then how does the model handle the increase of IC bins? One way to handle the stochastic ice nucleation process in a cloud parcel model is reported in Zhe et al. (2013).

Reference: Li, Zhe, Huiwen Xue, and Fan Yang. "A modeling study of ice formation affected by aerosols." Journal of Geophysical Research: Atmospheres 118, no. 19 (2013): 11-213.

3)      Please clearly describe how is the temperature perturbation linked to the vertical displacement of the cloud parcel related to its initial position. Do you need to assume a temperature lapse rate? If so, what is the value used in the model? Does the temperature perturbation also depend on ice microphysics, e.g., latent heat release during IC deposition? If not, please provide your justification.

4)      What are the conserved variables in the cloud parcel model? It seems that only the total water mixing ratio is conserved in the cloud parcel. If the temperature is not affected by microphysics, this cloud parcel model is not an “adiabatic cloud parcel model”.

5)      “The raw spectrum of vertical motions was generated by randomly selecting W from the observation.” It is not a surprise that the PDF of W agrees with the observation (Figure 1a), however, such random selection might lose the spatial and temporal correlations of W in nature. Please justify your method to generate the W series, especially, why it can represent the “uplift of moist air by wave motions on a variety of scales”.

6)      What is the growth equation of ice particles? Do you assume ice particles are spherical or nonspherical?

2.       Need some clarifications in Section 3.2

1)      “Therefore, only a small part of those samples (randomly selected) was used to create the plots below.” Please specify how many samples are used.

2)      The nice agreement between Obs and REF might also be related to the initial conditions you use in the model. In Figure 3 top three subplots, please add colored dots for 1000 observed samples that are used to initialize REF. In Figure 4, please add the relative frequency of NiRi from 1000 observed samples that are used to initialize REF.

Round 2

Reviewer 3 Report

The authors have addressed all my comments well. It is suitable for the publication in Atmosphere.